# From Hen Nutrition to Baking: Effects of Pomegranate Seed and Linseed Oils on Egg White Foam Stability and Sponge Cake Quality

**DOI:** 10.3390/foods14081417

**Published:** 2025-04-20

**Authors:** Marcin Lukasiewicz, Maja Dymińska-Czyż, Beata Szymczyk, Magdalena Franczyk-Żarów, Renata Kostogrys, Adam Florkiewicz, Paweł Ptaszek, Gabriela Zięć, Agnieszka Filipiak-Florkiewicz

**Affiliations:** 1Faculty of Food Technology, University of Agriculture in Krakow, al. Mickiewicza 21, 31-120 Krakow, Poland; maja.dyminska@o2.pl (M.D.-C.); magdalena.franczyk-zarow@urk.edu.pl (M.F.-Ż.); renata.kostogrys@urk.edu.pl (R.K.); adam.florkiewicz@urk.edu.pl (A.F.); pawel.ptaszek@urk.edu.pl (P.P.); gabriela.ziec@urk.edu.pl (G.Z.); agnieszka.filipiak-florkiewicz@urk.edu.pl (A.F.-F.); 2National Research Institute of Animal Production, ul. Krakowska 1, 32-083 Balice, Poland; beata.szymczyk@iz.edu.pl

**Keywords:** functional foods, pomegranate seed oil, eggs, unsaturated fatty acids, foams, sponge cakes

## Abstract

This study aimed to verify that enriching hens’ diets with pomegranate seed (PSO) and linseed oils (LSO) would maintain egg foaming and leavening capacity and improve the nutritional profile of egg-based products without compromising technological properties. It was shown in the previous studies that fortifying hen feed with PSO increased CLnA and CLA concentrations in raw eggs. In this study, two experiments with 25-week-old Hy-Line Brown laying hens have been carried out. Experiment 1 modified hens’ feed by incorporating PSO (0.5–1.5%) and 1.5% LSO. In Experiment 2, hens received feed containing PSO (0.5–1.5%). This research involved cake preparation, quality evaluation, and the assessment of egg white foam properties (stability, density, and gas bubble distribution). The chemical composition of sponge cake was determined. Results showed that PSO and LSO in hen feed enhanced egg leavening properties, while egg white-based foam properties matched the control group. The cakes showed improved health-promoting properties due to CLA and CLnA presence. The research confirmed that these beneficial acids were retained in the final sponge cake.

## 1. Introduction

Hen eggs have a low-energy value and are a rich source of nutrients. Their chemical composition is affected by both environmental conditions and genetic factors, as well as a hen’s diet, making them important for human nutrition [1,2].

The noteworthy high biological value of egg protein is attributed to its superiority over milk protein, making it a valuable supplement to plant and partially animal-based foods. Egg proteins boast excellent digestibility and assimilation properties [3,4]. Hen eggs contain approximately 9.7% fat, including 3% saturated, 4.3% monounsaturated, and 0.7% polyunsaturated fatty acids (PUFAs). The biological value of fats in the yolk is determined by the fatty acids present in them. Yolk lipids are composed of triacylglycerols (63.1%) and phospholipids (29.7%), with 6.2% of the yolk lipid mass consisting of free and esterified cholesterol. Oleic acid (C18:1 n-9) is the primary fatty acid in egg lipids, while linoleic acid (C18:2 n-6) and its derivative arachidonic acid (C20:4 n-6) are also relatively abundant. Additionally, yolk lipids contain small amounts of α-linolenic acid (C18:3 n-3) and its derivative docosahexaenoic acid (C22:6 n-3) [5,6].

Although the consumption of eggs varies significantly from country to country, they are commonly utilized as a raw material in the food technology sector, particularly in gastronomy and catering. Eggs are used in the preparation of a wide array of standalone dishes, ranging from snacks to breakfast, lunch, and dinner menu items. This is attributable to their properties, which include raising, solidifying, thickening, and emulsifying [7].

Egg whites possess various structural properties, such as gelling, coagulating, and binding water, which are essential for the preparation of numerous food items. Heating proteins cause structural alterations, contributing to the solidification and thickening of the products in which they are utilized. Additionally, the ability of egg whites to generate and maintain foam is crucial [8,9].

On the other hand, egg yolk is a functional additive in food production due to its excellent emulsifying properties, which make them an important ingredient in the production of mayonnaise, salad dressings, and cakes [10,11].

Following current dietary trends, consumers are seeking nutritious and well-balanced food options, leading to increased focus on eggs with enhanced quality. The contemporary consumer also shows interest in foods with functional properties [12]. Among these, eggs fortified with n-3 fatty acids, known as nutraceutical eggs, offer more than just nutritional value [13]. A regular intake of n-3 enriched eggs may help lower the likelihood of developing coronary heart disease, cognitive impairment, cancer, and neurodegenerative disorders [14].

Adjusting the composition of eggs for nutritional value has the potential to yield new products with increased health benefits. However, these modifications may also impact the processing and technological characteristics of eggs [15]. The primary areas of nutritional modification in eggs are related to yolk lipids, cholesterol, fatty acid composition, minerals, and vitamins [15,16]. Modifying the fatty acid profile of the yolk allows for an increase in the proportion of polyunsaturated fatty acids, while the proportion of saturated fatty acids remains unchanged [16].

Egg yolk lipids can be modified by feeding laying hens with an increasing proportion of plant-based oils [15,17]. The enrichment of eggs with PUFAs of the n-3 family can be achieved by providing hen feed containing flaxseed or flaxseed oil, which increases the α-linolenic acid and DHA contents of the eggs [18]. Another method is to add fish oils to hen feed, which increases the proportion of C22:6 acid (DHA) in yolk lipids and reduces the ratio of n-6 to n-3 acids. However, this method results in a change in the sensory characteristics of eggs, with a noticeable fishy aftertaste in the yolk [19].

Eggs enriched with flaxseed were found to have greater consumer acceptance due to the absence of a fishy smell [13].

Administering α-linolenic acid to the feed leads to its direct incorporation into the yolk lipid composition, with a disproportionately lower increase in the proportion of de-sired metabolites (DHA and EPA). Furthermore, the ratio of n-6 to n-3 acids in yolk lipids decreases due to the reducing proportions of oleic and arachidonic acids [20].

Pappas et al. investigated the possibility of enriching eggs with conjugated linoleic acid diene (CLA), which occurs naturally in foods of animal origin. With an increase in the content of these isomers in the laying hen feed, the CLA levels in the egg yolk lipids increased [21].

Nevertheless, the CLA-enriched eggs experienced adverse effects on their physicochemical characteristics, such as an increase in egg yolk firmness. Franczyk-Zarow et al. demonstrated that CLA influences the texture of egg yolks. Studies have shown that eggs enriched with CLA exhibit substantially greater hardness compared to regular eggs [22].

The negative impact of the modification of the fatty frication of eggs can manifest itself in different ways, including changes to the characteristics of both the yolk and egg white [23].

Our previous study on the fortification of hen feed with pomegranate seed oil as a source of punicic acid and the resulting changes in the chemical composition of raw eggs revealed that there was an increase in CLnA and CLA concentrations in egg yolk lipids with higher dietary CLnA levels in hens feed [2,16]. Our first study looked at enriching the diet with a relatively high addition of PSO in an amount equivalent to introducing CLnA into the hen’s diet at a level of 0.5–1.5% [2]. It is worth pointing out that this kind of modification results in an increase in the final product (egg) price. This can be a significant constraint on the marketability of the product. For this reason, recent studies have focused on obtaining a product (eggs) with a specific fatty acid profile and physicochemical properties, taking into account optimum economic solutions. This is the reason why replacing some share of pomegranate seed oil with linseed oil has been proposed. In these studies, the addition was 0.5% to 1.5% PSO rather than CLnA, respectively. We have stated that the inclusion of linseed and pomegranate seed oils did not alter the physical and chemical properties of raw eggs [16].

In the present work, we performed research on the technological properties of eggs enriched in pomegranate and linseed oil in complex food, including egg white-based foams and sponge cakes. The processing capabilities of fortified eggs, along with preserving their nutritional content during manufacturing, appear to be a crucial consideration. This is because consumers frequently eat egg-derived products, while the consumption of eggs as a standalone item is often restricted.

Based on this approach, we hypothesized that enriching hens’ diets with pomegranate seed and linseed oils will maintain or enhance the foaming capacity of egg whites and improve the nutritional profile of egg-based products like cakes, without compromising their technological properties. Moreover, CLA and CLnA fractions of eggs will be presented in the final product (e.g., cakes), which makes them a source of these valuable nutrients in the human diet. For a broader analysis of the technological properties of the modified eggs, the results obtained for raw material from groups in which feed was supplemented simultaneously with LSO and PSO were compared with those obtained from laying hens fed feed enriched only with PSO.

A comprehensive analysis of this product based on the presence of CLA and CLnA after egg processing appears to be essential for consumer benefits and has not yet been conducted or documented in the scientific literature.

## 2. Materials and Methods

### 2.1. Materials

#### 2.1.1. Design of Experiments

The eggs were gathered from two separate trials, which were carried out on Hy-Line Brown laying hens that were 25 weeks old, as described elsewhere [16]. These hens were kept in individual cages and had unrestricted access to water and food. Each bird was housed in a cage measuring 30 cm × 120 cm × 50 cm, providing a total floor area of 3600 cm^2^. From 17 to 24 weeks of age, the hens were fed a standard commercial diet during the pre-experimental phase. The experimental period lasted for 12 weeks, from the 25th to the 36th week of age, and was conducted under standard climatic conditions, including a temperature of 19 ± 2 °C, a relative humidity of 50–65%, and a 14 L:10 D light program. Water and food were provided ad libitum. The feed mixtures used in the experiments were carefully formulated and contained energy, proteins, and essential minerals to ensure optimal results. This approach aimed to prevent the overconsumption of feed due to compensatory mechanisms triggered by nutrient deficiencies. By meticulously formulating feed compositions, the study sought to achieve a balanced relationship between hen welfare and productivity, thus preventing reduced productivity due to overfeeding. The details of the feed composition and fatty acid profile of the oils used in all the experiments are provided in the Appendix A (Appendix A).

In both experiments, hens were randomly divided into four groups, each containing an equal number of 24 hens, and fed feed mixtures with varying proportions of linseed oil and pomegranate seed oil.

The experimental groups were as follows:

Experiment 1:

A—control group (rapeseed oil (RSO) 2.5%, linseed oil (LSO) 1.5%);

B—RSO 2.0%, LSO 1.5%, and pomegranate seed oil (PSO) 0.5%;

C—RSO 1.5%, LSO 1.5%, and PSO 1.0%;

D—RSO 1.0%, LSO 1.5%, and PSO 1.5%.

Experiment 2:

E—control group (RSO 4%);

F—RSO 3.5% and PSO 0.5%;

G—RSO 3.0% and PSO 1.0%;

H—RSO 2.5% and PSO 1.5%.

#### 2.1.2. Sampling Procedure

Twenty eggs randomly selected from each group were collected and broken. Yolks were separated from the albumen and the shell. An average laboratory sample was performed for eggs. Foams and cakes were performed in 3 replicates. All the chemical analyses were performed in triplicate.

### 2.2. Methods

#### 2.2.1. Foaming Properties

The preparation of the foam involved measuring 100 cm^3^ egg whites into an 800 cm^3^ beaker and weighing it to the nearest 0.01 g using a Mettler Toledo PB 602—S/FACT balance. The foam was then whipped for five minutes using a Zelmer mixer (Zelmer, Poland) set at 1400 rpm. The quality of the foam was assessed based on the following parameters:

Foaming capacity (ΔVP) was established based on [24] with some modification according to the following formula:$$\Delta \mathrm{VP}=\frac{m_{\mathrm{ew}\;}-{\;m}_{f}}{m_{f}}\cdot 100\%$$
where

m_ew_—the weight of 100 cm^3^ of egg white used in analysis;

m_f_—the weight of 100 cm^3^ of foam.

Foam stability (SP) was measured according to Ding et al. [24], with some modifications (stability was determined by the change in foam volume), as follows:$$\mathrm{SP}=\frac{V_{f}-{\;V}_{l30}}{V_{f}}\cdot 100\%$$
where

V_f_—the volume of foam;

V_l30_—the volume of the foam leakage after 30 min.

Foam index (IP) according to the following formula [25]:$$\mathrm{IP}=\frac{\Delta \mathrm{VP}}{100}\cdot \frac{\mathrm{SP}}{100}$$

Gas ratio (GR) is defined as foam fluffiness and an indirect indicator of foam texture, calculated as follows:$$\mathrm{GF}=\frac{V_{f}-V_{\mathrm{ew}}}{V_{f}}\cdot 100\%$$
where V_ew_ is the volume of egg white used in the analysis.

Foam density (GP) was calculated as follows:$$\mathrm{GP}=\frac{m_{f}}{V_{f}}$$

All the volumes were measured using graduated cylinder.

#### 2.2.2. Size Distribution of Gas Bubbles Suspended in Liquid

Analyses of the size distribution of air bubbles suspended in liquid were conducted using an inverted optical microscope. Images of the bubbles were captured with a digital camera and stored in the TIFF format (1280 × 1024 pixels). One thousand images were collected for each foam sample, resulting in a representative sample population of 25 × 104 bubbles [26]. The collected data allowed for a more precise description of the bubble size. The ImageJ (publication date of stable version: 2023.05.09; software version: 1.54e33) software [27] was used for image analysis with a self-developed macro constructed according to Labbafi et al. [28]. An automatic local threshold based on the Bernsen method was applied [29,30]. The data obtained pertain to the air bubbles. To convert the results to diameter, the area in pixels was calculated for a reference object, and the equivalent diameter (de) was calculated assuming the perfect circularity (Si—circle area) of the air bubbles [31]:$$d_{\mathrm{ei}}=\sqrt{\frac{{4S}_{i}}{\pi }}\;i=1,\ldots n$$

For each foam sample, a histogram of 5 × 104 was generated, and a consistent number of bins (25) was chosen across all the analyzed samples. As a result, five histograms were produced for each foam, enabling the extraction of dependable descriptive parameters. The Sauter mean diameter—d_32_—was calculated based on the following dependence (k = 2):$$d_{k+1,k}=\frac{\int_{0}^{+\infty }d_{e}^{k+1}\psi \left(d_{e}\right){\mathrm{dd}}_{e}}{\int_{0}^{+\infty }d_{e}^{k}\psi \left(d_{e}\right){\mathrm{dd}}_{e}}\approx \frac{\sum_{i=1}^{m}d_{\mathrm{ei}}^{k+1}\psi \left(d_{\mathrm{ei}}\right){\Delta d}_{e}}{\sum_{i=1}^{m}d_{\mathrm{ei}}^{k}\psi \left(d_{\mathrm{ei}}\right){\Delta d}_{e}},\;k\;\geq \;0;\;k\in N$$

The ψ(d) function present in the formula stands for empiric distributions obtained from the histograms.

The polydispersion coefficient was determined using the following correlation:$$\mathrm{PDI}=\frac{{\langle d\rangle }_{32}}{{\langle d\rangle }_{10}}$$

#### 2.2.3. Assessment of the Raising Properties of Eggs

Egg-based sponge cakes were produced using eggs from all the experimental groups, as follows [32]. Egg whites and yolks were separated after the eggs had been cracked open. The egg whites, which amounted to 100 g, were beaten using a kitchen blender (manufactured by Zelmer, Rzeszów, Poland) for one minute. Subsequently, 100 g of sugar (produced by Krakowska Spółka Cukrowa S.A., Toruń, Poland) and 70 g of egg yolk were added, and the mixture was blended for an additional minute. The flour mixture, which comprised 100 g of wheat flour (Polskie Zakłady Zbożowe w Krakowie S.A., Kraków, Poland) and the 2.2 g of baking powder (Food Care Sp. z o.o., Zabierzów, Poland), was then added to the mixture and mixed for 45 s. Rapeseed oil (11.5 g) (ZT Kruszwica S.A., Kruszwica, Poland) was then added to the batter and mixed for an additional 1.45 min. The batter was then poured into a 22.5 cm × 8.5 cm × 7.5 cm baking pan and baked in an electric oven that had been preheated to 200 °C for 16 min. After the cake had cooled, its quality was assessed.

The following parameters were determined:(a)Weight with an accuracy of 0.01 g using a Mettler Toledo PB 602—S/FACT balance (Columbus, OH, USA);(b)The volume of the cakes was determined using seed displacement by the method of Khna et al. with some modification [33];(c)Hardness of sponge cakes was conducted according to Rozyło and Laskowski with some modifications. TA.XT plus texture analyzer equipped with a 50 mm diameter cylindrical attachment (P-100) was used for measurement [34]. For the study, 2.5 cm thick cubes were prepared from each dough sample analyzed. The samples were subjected to double compression, with the roller travel speed set at 15 mm/min during the test. The compression process was carried out at a constant deformation of the samples equal to 50% of their height.

Additionally, an analysis of the basic chemical composition was done that included the following:(a)The Moisture of the cakes was established by drying the samples in a conventional oven at 98 °C for 24 h according to AOAC method [35];(b)The total fat content of cakes was assessed by the use of CO2 supercritical extraction: pump pressure—9000 PSI; cell temperature—100 °C; carbon dioxide flow rate—1.3 L/min; static time—5 min; and dynamic time—45 min (TFE2000 analyser, LECO, St. Joseph, MI, USA) [36];(c)Protein content was established based on the nitrogen amount that was analyzed using the TruSpec N LECO Company Analyzer. The analysis was done by means of the Dumas method, according to PN-EN ISO 16634-1:2008 [37], where values of N% were multiplied by 6.25 in order to calculate the protein %;(d)The ash content was analyzed by ashing the samples using a muffle furnace oven at 525 °C for 12 h [38];(e)The fatty acid analysis was performed stepwise. First, lipids were extracted from cakes by means of CO_2_ supercritical extraction (pump pressure—9000 PSI; cell temperature 100 °C; carbon dioxide flow rate—1.3 L/min; static time—5 min; and dynamic time—45 min) using FAT Ex-tractor TFE 2000 Leco, St. Joshep, MI, USA) [36]. After the extraction, lipids were methylated using sodium methylate [39]. In detail, 0.1 mL of extracted fat was placed in the glass test tube of 2 mL capacity, and 0.5 mL of 0.025 M of sodium methylate solution was added. The mixture was heated in a closed tube at 60 °C until the mixture was clear. The analysis of fatty acids was carried out using gas chromatography (Trace GC Ultra, Thermo Electron Corporation, Waltham, MA, USA). For the analysis, a Supelcowax 10 column (dimensions 30 m × 0.25 mm × 0.25 µm) was used. Helium was used as a sample carrier with a flow rate of 5 mL/min. The injector temperature was 220 °C. The temperature of the column was kept for 3 min at 60 °C, then increased at a rate of 7 °C/min up to 200 °C and then held at this temperature for 20 min. The detector temperature was set to 250 °C and the split flow was 10 mL/min. Peak identification was done using an external standard (FIM/FAME Supelco, Poznań, Poland). Fatty acid methyl esters were identified by comparing their retention times with authentic standards (Sigma Aldrich, Poznań, Poland) as well as the Punicic Acid Standard (Larodan Fine Chemicals AB, Malmö, Sweden).

All of the chemical analyses were conducted in triplicate.

#### 2.2.4. Statistical Analysis

The statistical analysis was performed using StatiStica v. 9.0 (Statsoft, Kraków, Poland) software. Mean values and standard deviations were calculated. The hypothesis of the normality of the distribution of results was verified using the Shapiro–Wilk test, and the homogeneity of variance was verified with the Lavene’s test. The results of the evaluation of the chemical compositions of sponge cakes as well as foam characteristic obtained in experiments 1 and 2 were subjected to one-way analysis of variance, and the significance of differences between the means (with a significance level of *p* ≤ 0.05) was determined using Duncan’s test. The results of the hardness of sponge cakes were subjected to the Kruskal–Wallis test. The significance of the differences is demonstrated by the *p*-values displayed in the Appendix A.

Moreover, to compare the results of the groups fed the feed with the same proportion of pomegranate seed oil from both experiments, the statistical analysis involved determining the significance of the differences between them using the Student’s *t*-test (significance level *p* < 0.05; Table 4 and Table 7).

The bubble distributions were assessed by the Kolmogorov–Smirnov test. The measurements were carried out in order to compare the distributions by pairs A:E, B:F, C:G, and D:H. Critical values [40] were determined using the application program written in Python (3.11.4 version release date: 6 June 2023). The Python language custom script was validated on synthetic data—random data with different probability densities (normal, lognormal, Weibull, and gamma).

## 3. Results

### 3.1. Egg Whites Foam Characteristics

To assess the foaming properties of eggs more comprehensively, a qualitative analysis of the foams was carried out. The foaming capacity and associated foam index of egg whites from Experiment 2 demonstrated considerable variation. The highest values of these parameters were discovered for the eggs belonging to the H group (obtained from hens fed 3.5% RSO and 1.5% PSO) (foaming capacity, 1005.76; foam index, 9.75). It was observed that the foams obtained exhibited varying stability (applicable to eggs from both experiments). The density of the analyzed foams was notably different depending on the type of eggs from Experiment 2 (as shown in Table 1).

Table 2 illustrates the mean gas bubble diameters in the generated foams and the average Sauter diameters of these foams. The data reveals that the average bubble diameters for Experiments 1 and 2 were significantly different from each other. On the other hand, the values of d_32_ are comparable. This can be attributed to the asymmetrical distribution of the gas bubble diameters in the foam and the substantial polydispersity of these diameters.

Figure 1 shows the distributions of the diameters of bubbles of gas dispersed in the liquid phase. It can be seen from the presented data that these distributions are similar in shape. Analysis using the Kolmogorov–Smirnov test showed that the distributions of bubble diameters of foams with B:F and C:G configurations are not significantly different from each other. In contrast, the diameter distributions for A:E and D:H foams showed significant differences in the distribution of bubble diameters.

### 3.2. Properties and Quality of Sponge Cakes

As shown in Table 3, the examination of the weights of sponge cakes obtained using eggs from both experiments revealed no appreciable variation (326.10 g to 338.00 g in Experiment 1 and 324.36 g to 332.04 g in Experiment 2). A similar observation was done in the case of the hardness of cakes obtained using eggs A to C. For cakes prepared using eggs D (hens feed containing RSO 1.0%, LSO 1.5% and PSO 1.5%), the hardness of the cakes was twice as high as in group A to C in Experiment 1. In the case of Experiment 2, the highest value of hardness was also indicated in the case of sponge cakes H; however, the increase in this parameter is much lower. It clearly shows that the highest share of PSO strongly influences the hardness parameter.

Additionally, significant variation in volume was detected in the cakes baked with eggs from Experiment 1. The significantly higher value of this parameter was attributed to cakes made with control eggs A (hens feed containing 2.5% RSO and 1.5% FSO) and D (hens feed containing RSO 1.0%, LSO 1.5% and PSO 1.5%) when compared to eggs B (RSO 2.0%, LSO 1.5% and PSO 0.5%) and C (RSO 1.5%, LSO 1.5% and PSO 1.0%). Such correlations were not observed in cakes made with eggs in Experiment 2, but it is worth noting that the volumes of cakes from Experiment 2 were significantly higher than those from Experiment 1.

Upon analyzing the physicochemical properties of the sponge cakes, it was determined that a 0.5% share of PSO in the laying hen feed, along with a constant quantity of LSO at 1.5%, resulted in a statistically significant reduction in the volume of the cakes when compared to the sponge cakes in group F, which were baked with eggs from hens not fed with LSO (Table 4). Similarly, when comparing cakes C and G, which were made with eggs from hens fed a diet containing 1% PSO, 1.5% LSO, and 3% RSO, respectively, a trend of reduced volume was observed as well.

The analysis of the basic chemical composition of sponge cakes showed significant variation according to the type of eggs used. A significantly lower dry matter content (65.43 g/100 g) was determined in cake G, concerning the results obtained for cakes E, F, and H (Table 5). Protein levels in cakes produced from eggs from both experiments were also significantly different. Sponge cakes containing control eggs from both experiments (A and E) contained more protein than those produced from eggs obtained from hens fed 0.5% pomegranate seed oil (B and F). When analyzing the fat content of the cakes, no clear trend was evident. In cakes baked with eggs obtained from Experiment 1, a reduction in total fat content was observed in cakes B, C, and D compared to control cake A. For Experiment 2, the lowest fat content was observed in cakes made with control eggs E. The ash content in the analyzed cakes was low and ranged in Experiment 1 from 1.25 g/100 g (cakes from eggs C) to 1.37 g/100 g (cakes from eggs D). In Experiment 2, as the amount of pomegranate seed oil in the eggs increased, the ash content decreased markedly in relation to the dough from control group E (1.35 g/100 g)—no pomegranate seed oil—and amounted to 1.20 g/100 g (dough H). However, no statistically significant differences were observed in these results.

The analysis of the fatty acid profile of the sponge cakes revealed that it varied significantly according to the type of eggs used in their production. The greater the addition of PSO to the laying feed (resulting in increased CLnA content in the eggs), the higher the level of conjugated trienes in the resulting cakes (Table 6). A similar relationship was also observed for CLA. The proportion of saturated, monounsaturated, and polyunsaturated acids in the product were also statistically significant. Notably, there was a significant increase in PUFAs in cakes baked on the basis of eggs obtained from hens fed 1% pomegranate seed oil, regardless of the presence of other oils in the laying feed.

The fatty acids composition revealed notable differences between B and F cakes, particularly with respect to polyunsaturated fatty acids, including C18:3, C18:2, and CLA (Table 7). A decrease in the aforementioned acids was observed in cakes made with eggs from hens fed diets containing 0.5% PSO and 1.5% LSO. C vs. G and D vs. H cakes displayed statistically significant variations between pairs.

Therefore, when 1% PSO, 1.5% LSO, and 1.5% RSO were added to the hens’ feed for cakes C, and 1% PSO and 3% RSO for cakes G, statistically significant results were obtained for C18:2, C18:3, and CLA acids compared to pair D vs. H, where eggs were obtained from hens fed feed containing 1.5% PSO, 1.5% LSO, and 1.0% RSO for cakes D, and 1.5% PSO and 2.5% RSO for cakes H, respectively. In each case, there was a statistically significant reduction in the content of the aforementioned acids in pairs D vs. H. The proportion of C:17:0 acid was significantly lower in cakes D and H, and polyunsaturated fatty acids were lower in cakes C vs. G. The addition of 1.5% PSO, 1.5% LSO, and 1.0% RSO to the laying hens’ feed in D doughs and 1.5% PSO and 2.5% RSO in H doughs significantly reduced the proportion of C18:2, C18:3, C17:0, and C22:0 acids and increased the proportion of C18:3 n-3 acid in C vs. G doughs.

## 4. Discussion

Eggs hold significant nutritional value for consumers and are equally important in food processing due to their technological properties. These properties include their capacity to emulsify water–oil systems, gelling characteristics, and their ability to form foam. Despite the extensive historical use of eggs in food production, challenges persist regarding emulsion stability and the impact of thermal treatment on both nutritional and technological properties. To date, these challenges have been addressed through biotechnological (enzymatic) processes, complexation, and the incorporation of low-molecular-weight substances, such as lecithin, among other methods [10]. In our previous study, it was shown that adding a pomegranate seed oil and linseed oil to hens’ feed did not have an effect on the physicochemical properties of raw eggs [16]. These findings allowed us to conclude that eggs with modified fatty acids profiles, including an increase in CLnA and CLA, obtained in Experiments 1 and 2, could serve as a valuable source of nutrients in the diet.

Foams are an essential component in the preparation of various food products, especially whipped creams, ice creams, cakes, sponge cakes, and meringues. In this group, cakes and cookies are particularly favored as snacks due to their widespread consumer appeal, convenience, and palatable taste [41]. The formation of foams is critical for achieving the desired sensory properties of foods [42,43]. Foamability and foam stability are pivotal topics in the study of food colloids, as they pertain to the dense arrangement of numerous bubbles that form an interconnected matrix in a wide array of food products, such as bread, ice cream, cakes, and mousse. As a result, these two indices—foamability and foam stability—are fundamental properties that influence the quality of foamed products. Additionally, these egg-related properties hold significance in other areas of the food industry, including dairy, meat, fats, and beverages [44].

In most cases, there was no change in the foaming properties of the egg white tested. The exceptions are the foams from eggs with the highest PSO content (D and H). The results of Experiment 1 revealed that foams produced from eggs differed significantly only in terms of stability, with the greatest stability observed in foams produced from eggs D, which were obtained from hens fed a diet containing 1.0% RSO, 1.5% LSO, and 1.5% PSO. In contrast, the foams produced from egg whites in Experiment 2 differed significantly in all the analyzed parameters. The highest foaming capacity was exhibited by H eggs, which were obtained from hens fed a diet containing the highest amount of PSO. These foams also had the highest foam index and percentage of gas in the foam. Meanwhile, the foam made from egg white G (feed contained 3% RSO and 1.0% PSO) showed the greatest stability. The highest density among the analyzed foams was observed in those made from egg whites E (feed with 4% RSO) and F (obtained from hens fed feed with 3.5% RSO and 0.5% PSO). The observed variations may be connected with the significantly higher protein content in egg whites from eggs from group D and H, as demonstrated in the previous publication [16]. Furthermore, as indicated by other researchers, the non-ideal separation of protein from yolk during egg processing, along with the indirect effects of the type of protein impurity, may influence the foaming process [44]. In the context of the foams examined in the experiments conducted, the differing yolk composition likely affects the nature of contamination and, consequently, the quality parameters of the protein foams. The analysis of the data reveals a significant difference in the average bubble diameters between Experiments 1 and 2. In contrast, the d_32_ values were similar. This observation can be explained by the uneven distribution of gas bubble diameters within the foam and the high degree of polydispersity present in these diameters.

Proteins are integral components in foam application products due to their remarkable functional properties, including foaming capability, emulsifying properties, and gel-formation ability. These attributes significantly contribute to the formation of smaller air cells, thereby enhancing foam stability [44]. The dimensions of the air cell significantly influence the texture of the final product. According to existing literature, larger air cells in crumbly baked goods may result in a more crumbly texture and the deformation of the product’s typical shape, both of which are considered undesirable quality attributes [45].

The preparation of cake, e.g., sponge cake, can serve as a method to assess the foaming capacity of egg white [46]. This is particularly significant among food foam products, as the protein foam is produced independently before being integrated with other ingredients. This process allows for the separate examination of the properties of both the foam and the final food product. Moreover, pastry products are popular due to their palatability, with sponge cakes being particularly esteemed among them. When evaluating confectionery products, consumers consider not only their health benefits, but also their texture. The quality of sponge cakes is largely influenced by their porosity [42]. The quality of these confectionery items depends on several factors, including the egg, which, when properly aerated, results in a cake with the desired porosity. Pycarell et al. have determined that the characteristics of egg whites are crucial for the quality of sponge cakes, but the properties of the egg yolk is also important [47]. It was demonstrated before by Kamat et al., who examined sponge cakes made from recipes in which the egg yolk was substituted with egg yolk plasma and/or granules [48]. The authors suggested that low-density lipoproteins assist aeration during mixing and that their functionality relies on their integrity.

When assessing sponge and fatty cake products, the specific gravity is a critical factor. The higher the specific gravity, the poorer the sensory characteristics of the product. The weight of the product is closely linked to its volume. The analysis of sponge cakes produced in Experiments 1 and 2 revealed no significant variation in their weight based on the type of eggs used. However, it is worth noting that the least dense sponge cakes were produced using eggs D and H, which were obtained from hens fed with feed containing the highest addition of PSO. The cakes produced using eggs from Experiment 1, on the other hand, varied in volume, with the lowest value observed in cakes made with eggs B and C (obtained from hens fed feed containing linseed oil and 0.5% or 1.0% of PSO, respectively). The reduced volume of the cake can be attributed to disruptions in gas retention, potentially caused by the reinforcement of bubble walls. This phenomenon has been documented in doughs with elevated fiber content [41]. However, in the cakes under investigation, this mechanism is not applicable, and the observed phenomenon is likely attributable to the interactions of yolk components (e.g., lipoproteins [47]), as well as the influence of heat on yolk components [42].

The examination of the hardness of baked sponge cakes showed a notable disparity dependent on the type of egg utilized. The trend has been observed that the higher the CLnA content in the cakes, the higher their hardness, but a significantly statistical difference in hardness was only found for the doughs with the highest CLnA concentration. Consequently, it can be inferred that the presence of CLnA in eggs does exert an impact on this parameter in sponge cakes. The basic chemical composition of the cakes exhibited considerable variation depending on the type of eggs used in their production. When it came to cakes baked using eggs from Experiment 2, the lowest (*p* < 0.05) level of dry matter was observed in cake G (1% of PSO). On the other hand, sponge cakes F (0.5% of PSO) possessed a lower dry matter content (*p* < 0.05) than sponge cakes E (control eggs in Experiment 2) and H (1.5% of PSO).

This study revealed substantial variation in the protein and fat contents of sponge cakes, depending on the type of eggs used. In particular, cakes baked using control eggs from Experiment 1, i.e., A, had significantly higher protein and fat contents than cakes B (0.5% of PSO) and D (1.5% of PSO). However, cakes baked with eggs from Experiment 2 (E—control) had significantly lower fat content than the others. The protein content of cakes baked with control eggs was similar to cake H (hens fed with feed containing 1.5% of PSO) and higher than cakes F (hens fed with feed containing 0.5% of PSO) and G (hens fed with feed containing 1.0% of PSO).

The examination of the fatty acids composition of sponge cakes demonstrated that as the ratio of conjugated trienes of linolenic acid and conjugated dienes of linoleic acid increased, the quantity of these components in cakes expanded as well. Moreover, an increase in the level of α-linolenic acid was detected in cakes produced from eggs obtained from hens fed flaxseed oil and 0.5% pomegranate seed oil. The fact that both CLnA and CLA were present in the final products indicates that they may be the source of these substances.

The superior quality and enhanced nutritional value of the resulting sponge cakes demonstrate the significant potential for egg fortification through the incorporation of PSO and LSO into hen feed. The findings of other authors suggest that, unlike the modifications of dough ingredients documented in the literature—such as the incorporation of oleogels [49], ground fruit peels (including pomegranates) [50], or linseed extracts [41]—the proposed alteration at the level of laying hen feed does not compromise dough quality. Moreover, it significantly enhances the nutritional and health benefits of the product, independent of its caloric content.

## 5. Conclusions

Emerging market trends and the growing interest in functional foods are creating new opportunities for food products. This is particularly relevant for eggs, which are a fundamental component of numerous popular diets worldwide. This study proved the hypothesis that enriching hen diets with pomegranate seed and linseed oils results in the production of eggs with maintained or enhanced foaming capacity (egg whites) and an improved nutritional profile of egg-based products, such as sponge cakes. Worth emphasizing in particular is that the investigation demonstrates conclusively that CLA and CLnA are present in the final products, specifically sponge cakes, following the thermal treatment required for their preparation.

The enriched eggs appear to have promising market potential, particularly when coupled with an effective information dissemination strategy. Their properties that are very close to unmodified “normal eggs” are crucial because consumer decisions regarding product consumption are heavily influenced by the amount of information available to them as well as by the similarity to the already-known product.

## Figures and Tables

**Figure 1 foods-14-01417-f001:**
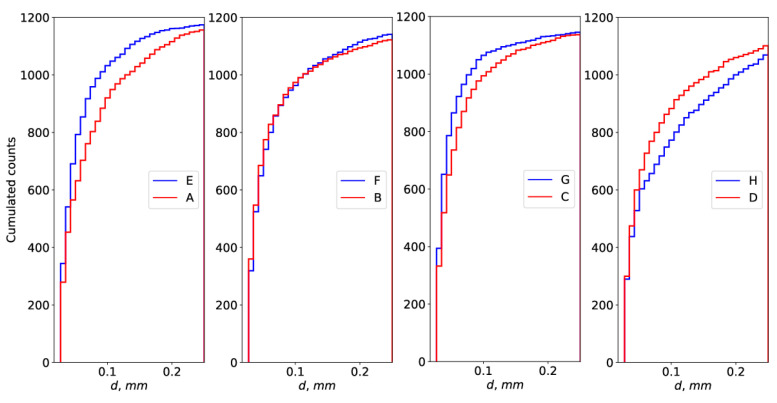
Distribution of bubble diameters of dispersed gas in the liquid phase.

**Table 1 foods-14-01417-t001:** Qualitative evaluation of foam.

Parameter	Experiment 1				Experiment 2			
	A	B	C	D	E	F	G	H
Foaming Capacity [%]	500.09 ^a^ ±10.04	553.85 ^a^ ±38.28	609.89 ^a^ ±13.51	540.07 ^a^ ±56.50	612.64 ^A^ ±38.31	590.02 ^A^ ±72.01	816.20 ^B^ ±91.15	1005.76 ^C^ ±11.83
Stability of foams [%]	94.41 ^a^ ±0.82	95.66 ^ab^ ±0.47	95.48 ^ab^ ±0.02	96.86 ^b^ ±0.27	95.35 ^A^ ±0.39	96.03 ^AB^ ±0.75	97.70 ^C^ ±0.05	97.00 ^B^ ±0.14
Foam index [%]	4.72 ^a^ ±0.04	5.29 ^a^ ±0.39	5.80 ^a^ ±0.98	5.23 ^a^ ±0.56	5.84 ^A^ ±0.38	5.66 ^A^ ±0.73	7.97 ^B^ ±0.88	9.75 ^C^ ±0.12
Percentage of gas in the foam [%]	83.33 ^a^ ±0.00	83.97 ^a^ ±0.90	85.00 ^a^ ±2.35	84.31 ^a^ ±1.38	85.49 ^AB^ ±0.59	84.52 ^A^ ±1.68	88.19 ^BC^ ±0.98	90.00 ^C^ ±0.00
Foam density [dm^−3^]	163.37 ^a^ ±1.77	155.66 ^a^ ±13.22	145.14 ^a^ ±21.34	148.76 ^a^ ±19.88	141.75 ^B^ ±9.04	148.46 ^B^ ±10.65	111.37 ^A^ ±7.57	90.18 ^A^ ±0.16

Values marked with different letters in experiment: 1 (^a,b^) and 2 (^A,B,C^) differ significantly *p* ≤ 0.05.

**Table 2 foods-14-01417-t002:** Calculated foam parameters.

Parameter	Experiment 1				Experiment 2			
	A	B	C	D	E	F	G	H
d_e_ [mm]	0.082	0.083	0.079	0.098	0.065	0.078	0.070	0.072
d_32_ [mm]	0.187	0.187	0.187	0.188	0.187	0.187	0.187	0.186
PDI	0.44	0.45	0.42	0.52	0.35	0.42	0.37	0.39

d_e_—equivalent diameter; d_32_—the Sauter mean diameter; and PDI—polydispersity coefficient.

**Table 3 foods-14-01417-t003:** Physicochemical properties of sponge cakes.

Parameter	Experiment 1				Experiment 2			
	A	B	C	D	E	F	G	H
Cake weight [g]	338.00 ^a^ ±4.15	334.92 ^a^ ±4.72	337.69 ^a^ ±5.52	326.10 ^a^ ±7.07	329.12 ^A^ ±10.28	324.36 ^A^ ±11.59	332.04 ^A^ ±1.63	326.38 ^A^ ±2.87
Cake volume [cm^3^]	1075.00 ^b^ ±106.06	835.00 ^a^ ±35.35	815.00 ^a^ ±49.49	1091.50 ^b^ ±4.94	1000.00 ^A^ ±155.56	1075.00 ^A^ ±7.07	1015.00 ^A^ ±21.21	1055.00 ^A^ ±21.21
Hardness [N]	387.85 ^a^ ±76.618	406.07 ^a^ ±123.21	522.36 ^a^ ±117.19	1032.08 ^b^ ±189.19	392.52 ^A^ ±57.59	295.29 ^A^ ±136.30	478.17 ^A^ ±51.40	679.06 ^A^ ±171.84

Values marked with different letters in experiment: 1 (^a,b^) and 2 (^A^) differ significantly *p* ≤ 0.05.

**Table 4 foods-14-01417-t004:** Comparison of physicochemical properties of sponge cakes obtained in Experiments 1 and 2—*p*-values (Student’s *t*-test; significance level *p* < 0.05).

Parameter	Sponge Cakes			
	A vs. E	B vs. F	C vs. G	D vs. H
Cake weight [g]	0.341	0.145	0.061	0.962
Volume of the cake [cm^3^]	0.629	**0.011**	**0.034**	0.141
Hardness [N]	0.557	0.850	0.137	0.202

The bold values designates the statistical significance between groups.

**Table 5 foods-14-01417-t005:** Analysis of basic chemical composition of sponge cakes [g/100 g d.m.].

Parameter	Experiment 1				Experiment 2			
	A	B	C	D	E	F	G	H
Dry matter	68.51 ^a^ ±0.65	64.60 ^a^ ±1.14	67.21 ^a^ ±2.52	66.33 ^a^ ±0.75	69.20 ^C^ ±1.22	66.27 ^B^ ±1.21	65.43 ^A^ ±0.32	67.69 ^C^ ±0.28
Protein	13.49 ^c^ ±0.10	13.34 ^b^ ±0.03	13.39 ^bc^ ±0.02	13.12 ^a^ ±0.01	13.63 ^B^ ±0.10	13.11 ^A^ ±0.11	13.31 ^A^ ±0.09	13.75 ^B^ ±0.11
Fat	12.05 ^c^ ±0.07	11.25 ^b^ ±0.07	10.85 ^b^ ±0.35	10.30 ^a^ ±0.00	9.97 ^A^ ±0.04	11.30 ^B^ ±0.14	11.10 ^B^ ±0.14	12.25 ^C^ ±0.21
Ash	1.33 ^a^ ±0.00	1.27 ^a^ ±0.00	1.25 ^a^ ±0.03	1.37 ^a^ ±0.00	1.35 ^A^ ±0.03	1.33 ^A^ ±0.01	1.23 ^A^ ±0.01	1.20 ^A^ ±0.01

Values marked with different letters in Experiment 1 (^a,b,c^) and 2 (^A,B,C^) differ significantly *p* ≤ 0.05.

**Table 6 foods-14-01417-t006:** Fatty acid profile of cakes in relation to the presence of linseed oil and the proportion of pomegranate seed oil in the feed of hens [%].

Acid	Experiment 1				Experiment 2			
	A	B	C	D	E	F	G	H
Saturated Fatty Acids								
C14:0 tetradecanoic (myristic acid)	0.46 ^b^ ±0.03	0.39 ^a^ ±0.06	0.45 ^b^ ±0.02	0.42 ^ab^ ±0.00	0.52 ^A^ ±0.02	0.48 ^A^ ±0.02	0.47 ^A^ ±0.01	0.49 ^A^ ±0.01
C15:0 pentadecanoic (pentadecylic acid)	0.09 ^a^ ±0.00	0.07 ^a^ ±0.00	0.09 ^a^ ±0.00	0.09 ^a^ ±0.00	0.11 ^B^ ±0.01	0.08 ^A^ ±0.00	0.09 ^A^ ±0.01	0.09 ^A^ ±0.01
C16:0 hexadecanoic (palmitic acid)	16.85 ^a^ ±0.47	17.59 ^b^ ±0.04	17.17 ^b^ ±0.35	16.94 ^a^ ±0.02	16.65 ^A^ ±0.13	17.10 ^A^ ±0.33	16.74 ^A^ ±0.18	17.11 ^A^ ±0.00
C17:0 heptadecanoic (margaric acid)	0.21 ^a^ ±0.02	0.15 ^a^ ±0.00	0.19 ^a^ ±0.00	0.24 ^b^ ±0.01	0.23 ^A^ ±0.04	0.23 ^A^ ±0.05	0.24 ^A^ ±0.01	0.25 ^A^ ±0.00
C18:0 octadecanoic (stearic acid)	7.73 ^a^ ±0.61	7.90 ^a^ ±0.39	7.7 ^a^ ±0.67	7.95 ^a^ ±0.41	6.21 ^A^ ±0.22	8.07 ^B^ ±0.21	6.93 ^A^ ±0.74	7.59 ^B^ ±0.07
C20:0 eicosanoic acid (arachidic acid)	0.33 ^a^ ±0.07	0.3 ^a^ ±0.06	0.32 ^a^ ±0.03	0.39 ^a^ ±0.01	0.38 ^A^ ±0.07	0.36 ^A^ ±0.00	0.36 ^A^ ±0.00	0.38 ^A^ ±0.01
C22:0 docosanoic (behenic acid)	0.08 ^b^ ±0.01	0.00 ^a^ ±0.00	0.08 ^b^ ±0.00	0.12 ^b^ ±0.00	0.05 ^A^ ±0.07	0.11 ^B^ ±0.01	0.10 ^B^ ±0.00	0.13 ^B^ ±0.00
Monounsaturated Fatty Acids								
C14:1 9-tetradecenoic (myristoleic acid)	0.05 ^a^ ±0.00	0.04 ^a^ ±0.02	0.05 ^a^ ±0.01	0.05 ^a^ ±0.00	0.07 ^B^ ±0.00	0.06 ^A^ ±0.00	0.06 ^A^ ±0.00	0.06 ^A^ ±0.00
C16:1 *trans*-3-hexadecenoic	1.15 ^a^ ±0.07	1.10 ^a^ ±0.07	1.0 ^a^ ±0.05	0.9 ^a^ ±0.02	1.45 ^B^ ±0.05	1.16 ^B^ ±0.09	1.23 ^AB^ ±0.00	1.08 ^A^ ±0.00
C16:1 9-cis-Hexadecenoic (palimitoleic acid)	4.06 ^b^ ±0.51	2.86 ^a^ ±0.15	2.84 ^a^ ±0.34	3.05 ^a^ ±0.04	3.86 ^B^ ±0.05	3.32 ^A^ ±0.21	2.84 ^A^ ±0.34	2.97 ^A^ ±0.01
C17:1 10-heptadecenoic acid	0.22 ^a^ ±0.07	0.19 ^a^ ±0.02	0.17 ^a^ ±0.00	0.22 ^a^ ±0.04	0.25 ^A^ ±0.07	0.19 ^A^ ±0.02	0.21 ^A^ ±0.03	0.21 ^A^ ±0.01
C18:1 *cis*-9-Octadecenoic (oleic acid)	42.36 ^a^ ±0.95	43.09 ^a^ ±0.16	41.76 ^a^ ±0.76	39.94 ^a^ ±0.40	40.18 ^A^ ±0.91	41.80 ^A^ ±0.18	42.72 ^A^ ±1.01	41.08 ^B^ ±0.26
C22:1 (13Z)-docos-13-enoic (erucic acid)	0.12 ^b^ ±0.04	0.00 ^a^ ±0.00	0.13 ^b^ ±0.01	0.14 ^b^ ±0.01	0.12 ^A^ ±0.04	0.13 ^A^ ±0.00	0.13 ^A^ ±0.00	0.14 ^A^ ±0.02
Polyunsaturated Fatty Acids								
C16:2 Hexadecadienoic acid	0.08 ^a^ ±0.00	0.08 ^a^ ±0.00	0.08 ^a^ ±0.00	0.09 ^b^ ±0.01	0.09 ^B^ ±0.00	0.09 ^B^ ±0.01	0.09 ^B^ ±0.00	0.08 ^A^ ±0.01
C18:2 *n*−6 *cis,cis*-9,12-octadecadienoic (linoleic acid)	16.75 ^a^ ±0.00	16.63 ^a^ ±0.01	16.92 ^a^ ±0.07	16.81 ^a^ ±0.12	17.36 ^B^ ±0.22	16.55 ^A^ ±0.12	16.8 ^A^ ±0.19	16.67 ^A^ ±0.06
C18:2 Conjugated linoleic acids—CLA	0.12 ^a^ ±0.00	0.74 ^b^ ±0.01	1.49 ^c^ ±0.04	2.23 ^b^ ±0.01	0.10 ^A^ ±0.02	1.21 ^B^ ±0.02	1.66 ^C^ ±0.09	2.21 ^D^ ±0.02
C18:3 *n*-3 *cis,cis,cis*-9,12,15-octadecatrienoic (α-linolenic acid)	8.08 ^b^ ±0.41	7.31 ^a^ ±0.19	7.75 ^b^ ±0.02	7.86 ^ab^ ±0.17	7.44 ^B^ ±0.55	7.17 ^A^ ±0.35	7.37 ^A^ ±0.21	6.84 ^A^ ±0.07
C18:3 Conjugated linolenic acid - CLnA	0.00 ^a^ ±0.00	0.23 ^b^ ±0.08	0.46 ^c^ ±0.35	0.84 ^d^ ±0.04	0.00 ^A^ ±0.00	0.32 ^B^ ±0.06	0.62 ^C^ ±0.01	0.97 ^D^ ±0.02
C20:2 eicosadienoic	0.07 ^a^ ±0.02	0.06 ^a^ ±0.02	0.07 ^a^ ±0.01	0.08 ^a^ ±0.01	0.06 ^A^ ±0.02	0.08 ^A^ ±0.01	0.08 ^A^ ±0.00	0.09 ^A^ ±0.01
C20:3 *n*-6 *cis,cis,cis*-8,11,14-eicosatrienoic dihomo-γ-linolenic acid	0.05 ^a^ ±0.00	0.10 ^b^ ±0.02	0.02 ^a^ ±0.02	0.05 ^a^ ±0.01	0.06 ^A^ ±0.02	0.05 ^A^ ±0.01	0.04 ^A^ ±0.01	0.06 ^A^ ±0.00
C20:4 *n*-6 5,8,11,14-*all-cis*-eicosatetraenoic (arachidonic acid)	0.43 ^ab^ ±0.15	0.39 ^b^ ±0.01	0.33 ^a^ ±0.19	0.42 ^ab^ ±0.06	0.17 ^A^ ±0.03	0.57 ^B^ ±0.12	0.34 ^B^ ±0.24	0.52 ^B^ ±0.03
C22:6 *n*-3 docosahexaenoic —DHA (cervonic acid)	0.21 ^b^ ±0.06	0.20 ^b^ ±0.02	0.18 ^b^ ±0.11	0.21 ^b^ ±0.05	0.00 ^A^ ±0.00	0.17 ^B^ ±0.03	0.08 ^A^ ±0.07	0.14 ^B^ ±0.02
Other C18:2, C18:3, CLA	0.45 ^a^ ±0.07	0.53 ^a^ ±0.05	0.64 ^b^ ±0.00	0.85 ^b^ ±0.07	0.59 ^B^ ±0.06	0.69 ^A^ ±0.03	0.77 ^B^ ±0.04	0.82 ^B^ ±0.00
Saturated fatty acids—SFA (%)	25.77 ^a^ ±0.92	26.42 ^a^ ±0.31	26.03 ^a^ ±0.96	26.17 ^a^ ±0.41	24.16 ^A^ ±0.34	26.41 ^A^ ±0.60	24.9 ^A^ ±0.92	26.03 ^A^ ±0.09
Monounsaturated fatty acids— MUFA (%)	47.98 ^b^ ±0.63	47.29 ^b^ ±0.11	46.01 ^a^ ±1.19	44.38 ^a^ ±0.26	49.94 ^B^ ±0.67	46.66 ^B^ ±0.53	47.20 ^B^ ±0.71	45.54 ^A^ ±0.28
Polyunsaturated fatty acids— PUFA (%)	26.24 ^a^ ±0.28	26.28 ^a^ ±0.21	27.95 ^b^ ±0.23	29.45 ^b^ ±0.14	25.89 ^A^ ±0.32	26.92 ^A^ ±0.07	27.86 ^B^ ±0.21	28.42 ^C^ ±0.19

Values marked with different letters in experiment: 1 (^a,b,c^) and 2 (^A,B,C^) differ significantly *p* ≤ 0.055.

**Table 7 foods-14-01417-t007:** Comparison of physicochemical characteristics and fatty acid profile of cakes obtained in Experiment 1 and 2—*p*-values (Student’s *t*-test).

Parameter	Sponge Cake			
	A vs. E	B vs. F	C vs. G	D vs. H
PUFA				
C16:2	0.422	0.422	**0.000**	0.422
C18:2 n-6	0.095	0.148	0.360	0.067
C18:2	**0.000**	**0.000**	**0.006**	**0.003**
C18:3 n-3	0.417	0.141	0.785	**0.003**
C18:3	**0.001**	**0.018**	**0.023**	**0.003**
C20:2 n-9	0.542	0.591	0.422	0.051
C20:3 n-6	0.830	0.422	0.062	0.183
C20:4 n-6	0.039	0.406	0.821	0.300
C22:6 n-3	**0.034**	0.518	0.138	0.709
Other C18:2, C18:3, CLA	0.063	0.048	**0.040**	**0.000**
MUFA				
C14:1	**0.037**	0.422	0.422	0.422
C16:1	**0.007**	0.918	0.121	0.666
C16:1	**0.004**	0.203	0.960	0.660
C17:1	0.690	0.633	0.516	0.057
C18:1	**0.026**	0.497	0.658	0.356
C22:1	0.591	0.899	**0.000**	0.492
SFA				
C14:0	**0.019**	0.685	0.245	0.191
C15:0	0.311	0.292	0.311	0.698
C16:0	0.094	0.586	**0.023**	0.832
C17:0	0.903	0.759	**0.016**	**0.008**
C18:0	**0.035**	0.529	0.246	0.818
C20:0	0.840	0.751	0.384	0.142
C22:0	0.332	0.167	**0.002**	**0.012**
Share of PUFA	**0.004**	0.085	**0.017**	0.164
Share of MUFA	**0.008**	0.153	0.875	0.642
Share of SFA	**0.034**	0.498	0.164	0.994

The bold values designates the statistical significance between groups.

## Data Availability

The original contributions presented in the study are included in the article/Appendix A. Further inquiries can be directed to the corresponding author.

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
