# Peer review of "From Hen Nutrition to Baking: Effects of Pomegranate Seed and Linseed Oils on Egg White Foam Stability and Sponge Cake Quality"

_foods, 2025, doi:10.3390/foods14081417_

Round 1
Reviewer 1 Report
Comments and Suggestions for Authors
Generally, the introduction is unnecessarily long; it should be significantly shortened to highlight the key points.
P1, L29-31: The chemical composition of eggs is not only affected by environmental conditions and genetic factors, but also by hen diet.
P2, L77-79: The statement “Modifying the fatty acid profile of the yolk allows for an increase in the proportion of monounsaturated and polyunsaturated fatty acids, while the proportion of saturated fatty acids remains unchanged” is incorrect. Adjusting the levels of MUFA and PUFA would inevitably alter the proportion of SFA as well.
P2, L80-81: You stated “Egg yolk lipids can be modified by feeding laying hens with an increasing proportion of plant based oils, which raises the content of oleic acid (C18:1) in these fraction.” However, the fatty acid composition of egg yolk depends on the specific type of vegetable oil used in the hen's diet. While oils rich in oleic acid (e.g., olive oil) will increase its content, other oils, such as soybean or flaxseed oil, can lead to a higher proportion of linoleic acid (C18:2) or α-linolenic acid (C18:3). Therefore, the modification of yolk lipids is not limited to oleic acid alone but varies depending on the dietary oil source.
P3, L111: Provide a reference after “Our previous study”.
P4, Materials: It is not clear why the experiment is divided into two parts, what was the goal? That should be clarified. The chemical composition of the control diet is shown in the supplementary material; it should be noted that this is the control, and the raw materials should be arranged in descending order of content.
P5, L191: You stated “Foam stability (SP) was measured according to Ding et. al [24] with some modifications”. What modifications were made?
P5, L201: How did you determine Vev?
In some parts of the text, references are not provided; this should be checked and corrected (L316, L319, L335, L358, L367, L386, L398).
Table 1. The units for the parameters are missing from the table.
Table 2. Please provide statistical significance for the parameters. Also, specify what the abbreviations in the table mean.
Figure 1: The quality of Figure 1 is not clear, and it is difficult to determine which samples are being referenced. Additionally, the labels for the x and y axes are not clear.
P8, L334: Modify the title of 3.2. Loosening properties off eggs, the case of sponge cakes
P14, L447: You stated “Research claims that the..” Please provide appropriate references.
P14, L454-455: You stated “Consequently, it can be inferred that the presence of CLnA in eggs does exert a substantial impact on this parameter in sponge cakes”. However, based on the results presented in Table 3, Group D exhibited higher hardness compared to the other groups in the experiment, which contradicts this inference. This discrepancy should be addressed and clarified.
In the discussion section, there is a repetitive presentation of the results, while a more in-depth and substantive analysis of the findings is lacking. A stronger focus on interpreting and contextualizing the results would greatly enhance the quality of the manuscript. The discussion section has to be improved.
Author Response
We would like to thank the Reviewer for the valuable comments to our manuscript submitted to Foods and entitled “Incorporation of Pomegranate Seed and Linseed Oils in Laying Hen Diets: Consequences for Egg White Foam Stability and Cake Quality”.
Ale the changes made were highlighted in green yellow in the file containing revised manuscript. We agree with all the suggestions made by the reviewer and bellow we would like to shown the detailed changes in the clear tabular form in a file attached.

Reviewer 2 Report
Comments and Suggestions for Authors
Please see in the attach.

Author Response

(The authors gave the same response as above.)

Reviewer 3 Report
Comments and Suggestions for Authors
Dear Authors,
My comments are attached.

The English language needs editing.
Author Response

(The authors gave the same response as above.)

Round 2
Reviewer 1 Report
Comments and Suggestions for Authors
The authors have addressed the suggestions and made the necessary corrections to improve the quality of the manuscript.
Author Response
We would like to thank the Reviewer for the valuable comments to our manuscript submitted to Foods and entitled “Incorporation of Pomegranate Seed and Linseed Oils in Laying Hen Diets: Consequences for Egg White Foam Stability and Cake Quality”. We are happy to fulfill all reviewers suggestions.
Reviewer 3 Report
Comments and Suggestions for Authors
Dear Authors,
Thank you for the revised paper, which has been improved. However, some concerns are:
- The Abstract should be better written. It isn't clear when you refer to Experiment 2 before Experiment 1. The aim is introduced, but the specific setup and variables (PSO vs. LSO, control groups, concentrations) are unclear or logically introduced. The conclusion mentions CLA and CLnA without explaining them earlier in the Abstract. It ends by stating that the hypothesis was confirmed but never clearly states the hypothesis in the first place. Furthermore, the Abstract lacks basic details like the number of replicates, birds, or cakes tested.
- Lines 114-134 (the "research gap") are confusing. First, they discuss earlier results with PSO, then introduce LSO without clearly linking the transition. The "gap" loosely implies that "eggs enriched with CLnA are more expensive, and people often don't eat eggs alone." It ends with "this work is the next stage…" but doesn't clearly state what exactly is new about this study. Line 121 mentioned "recent studies" but with only one cited study (number 2).
- Conclusion: it is difficult to follow due to poor grammar, weak structure, and awkward expressions. "It enhances their leavening properties…" → is "it" the egg, the fatty acids, or the oil? Needs clarification. Keep the roles of eggs (as ingredients) vs. the end products (cakes) distinct and well-explained.
The English language needs editing.
Author Response
We would like to thank the Reviewer for the valuable comments to our manuscript submitted to Foods and entitled “Incorporation of Pomegranate Seed and Linseed Oils in Laying Hen Diets: Consequences for Egg White Foam Stability and Cake Quality”.
Ale the changes made were highlighted in yellow in the file containing revised manuscript. We agree with all the suggestions made by the reviewer and bellow we would like to shown the detailed changes in the clear tabular form.
